# Comparative Fruit Morphology and Anatomy of Wild Relatives of Carrot (*Daucus*, Apiaceae)

Dariusz Kadluczka and Ewa Grzebelus *

Department of Plant Biology and Biotechnology, Faculty of Biotechnology and Horticulture,
University of Agriculture in Krakow, Al. Mickiewicza 21, 31-120 Krakow, Poland
* Correspondence: ewa.grzebelus@urk.edu.pl

**Abstract:** Fruit morphological and anatomical characteristics are essential in the taxonomy of Apiaceae. *Daucus* L. is one of the most important genera of the family Apiaceae, as it contains the cultivated carrot, a crop of great economic importance, and about 40 wild species that could serve as potential sources of genetic diversity for crop improvement. However, the taxonomic and phylogenetic relationships among these species have not yet been fully clarified. In this study, we comparatively investigated the fruit morphology and anatomy of 13 *Daucus* taxa and four closely related non-*Daucus* species using light and scanning electron microscopy to evaluate the taxonomic value of these characteristics. A wide range of variations was observed in the fruit morpho-anatomical characteristics across the taxa and revealed several diagnostically valuable features, thus proving to be taxonomically useful. For *Daucus*, the observed differences included the fruit size (2.1–8.4 mm), shape (from ellipsoid to oblong), and weight (0.079–1.349 g/100 fruits), as well as the fruit surface sculpturing and some anatomical characteristics, i.e., the presence/absence and size of vittae, the shape and size of vascular bundles, and the shape of exocarp cells. This study contributes to a better understanding of the relationships among the genus *Daucus*.

**Keywords:** Apioideae; carpology; crop wild relatives; Daucinae; mericarp; plant systematics; schizocarp; Torilidinae



## 1. Introduction

The genus *Daucus* L. belongs to the large and complex family Apiaceae, which comprises approximately 3820 species in 466 genera that are widely distributed all around the world, especially in the temperate regions of Eurasia and North America [1]. This cosmopolitan family is considered one of the most economically important families, and it includes a number of food crops, herbs, and spices [2]. *Daucus* contains carrot (*D. carota* subsp. *sativus* Hoffm.), the only cultivated member of the genus, which is a crop of great importance for human nutrition as it serves as a major source of α- and β-carotene (vitamin A precursors) in the diet [3,4]. The taxonomic and phylogenetic relationships among *Daucus* species have not yet been fully clarified. Traditionally, the genus comprised 20–25 species, as inferred from morphological and anatomical data [2,5]. However, recent studies based on different molecular data have led to a better understanding of the species boundaries and phylogenetic relationships among *Daucus* and its close relatives in the Apioideae subfamily [6–13]. Following these revisions, the genus has been extended to include nine other genera, and it now contains about 40 species positioned in two main clades [11].

The wild species of *Daucus* are widespread in the temperate parts of the Northern Hemisphere, most commonly in the Mediterranean region, which is considered the center of this genus's diversity; however, few species occur in South America, Australia, and tropical Africa [14,15]. They are mostly herbaceous biennials, rarely annuals [14], but a few rosette treelets (endemic to Macaronesia) also exist [16]. Most *Daucus* species are diploids with chromosome numbers of 2*n* = 16, 18, 20, or 22; however, some tetra- and hexaploids

have also been reported [15,17,18]. Regarding the genome size within the genus, nuclear DNA content estimates based on flow cytometry are available for several wild species and subspecies, as well as for many cultivated carrots, ranging from 0.920 to 3.228 pg/2C DNA [19–21].

The fruit of Apiaceae is typically a schizocarp that splits at maturity into two—usually equal—ribbed, one-seeded mericarps. Each mericarp has five primary ribs: three dorsal (one median and two lateral) and two marginal (closest to the commissure), which are separated by furrows (valleculae). The primary ribs enclose one or more vascular bundles that are often associated with schizogenous secretory canals (rib oil ducts). Another set of secretory canals (vittae) are located in the valleculae and the commissural area but are not associated with the vasculature. In some groups, secondary ribs develop from the valleculae, and they have no vascular bundles (see Figure 1) [1,22,23]. The fruit's structural characteristics, especially the number and distribution of vittae and vascular bundles, as well as rib/wing morphology, have proven to be exceptionally useful for the taxonomy of Apiaceae (e.g., [24–32]). Regarding *Daucus*, several decades ago, Sáenz Laín [5] published a taxonomic monograph of the genus based on morpho-anatomical analyses, providing some observations of the fruit morphology and anatomy of *Daucus* taxa; however, this was a largely intuitive classification that did not contain specimen citations, detailed descriptions, or distribution maps [33]. More recently, Mezghani et al. [34] studied the patterns of phenotypic diversity of fruits among Tunisian *Daucus* germplasm collection, whereas Wojewódzka et al. [35] investigated fruit evolution in many members of the Apiaceae tribe Scandiceae, including some *Daucus* taxa, and outgroups to assess adaptive shifts associated with the evolutionary switches between anemochory and epizoochory, as well as to identify possible dispersal syndromes.

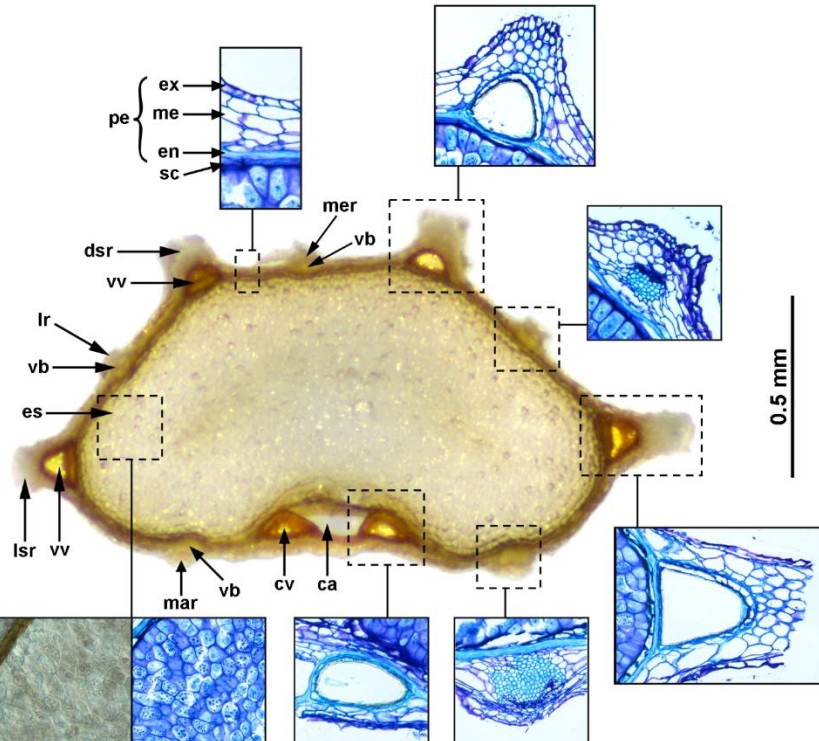

**Figure 1.** Transverse section of a mericarp of *Daucus* sp., indicating the anatomical structures considered in this study and their terminology. The insets show the corresponding structures, as seen by light microscopy. Abbreviations: ca, cavity; cv, commissural vitta; dsr, dorsal secondary rib; en, endocarp; es, endosperm; ex, exocarp; lr, lateral primary rib; lsr, lateral secondary rib; mar, marginal primary rib; me, mesocarp; mer, median primary rib; pe, pericarp; sc, seed coat; vb, vascular bundle; vv, vallecular vitta.

To address the rising need for food and to ensure food security for a constantly growing population, plant breeders require access to new genetic resources that could be used in crop breeding programs to expand the genetic variation of crops that has been lost during domestication. Such a large pool of genetic diversity can be found in crop wild relatives, which—due to their high adaptability to a wide range of habitats and environmental conditions—represent an important reservoir of agronomically important genes [36–38]. In this context, wild *Daucus* species may play a crucial role in the process of improving modern agriculture, being a valuable source of genes potentially useful for breeding purposes, e.g., for producing new crop varieties that could be disease-resistant, tolerant to abiotic stress, higher-yielding, male-sterile, or more nutritious [9,14]. In light of this, a better understanding of species relationships within the genus *Daucus* may greatly contribute to the development of future carrot breeding programs.

Given the significance of wild *Daucus* species and the great economic importance of the cultivated carrot, as well as the taxonomical usefulness of fruit structural characteristics in Apiaceae, this study aimed to compare the fruit morphology and anatomy of *Daucus* taxa using light and scanning electron microscopy (SEM) and to evaluate the diagnostic value of these characteristics. In this study, which is a continuation of our previous work [21], we selected a representative sample that covered the two main *Daucus* subclades (13 taxa) and four closely related non-*Daucus* species. We used the same accessions that have commonly been used in previous phylogenetic and (cyto)taxonomic research on the genus [9,12,13,21,39].

## 2. Materials and Methods

### 2.1. Plant Material

The study materials were ripe fruits (mericarps) of 13 *Daucus* taxa (14 accessions) and four closely related non-*Daucus* species (outgroup). The *Daucus* accessions comprised 12 wild taxa belonging to *Daucus* subclades I and II, as well as two cultivated carrots. The fruit samples of wild *Daucus* and non-*Daucus* accessions were provided by the USDA-ARS North Central Regional Plant Introduction Station (Ames, IA, USA), whereas the fruits of the carrot accessions were either purchased from commercial sources or obtained from the collections of the Department of Plant Biology and Biotechnology, University of Agriculture in Krakow (Krakow, Poland). The following taxa were used (chromosome numbers [17,18] and accession numbers [PI = USDA Plant Introduction numbers] are given in brackets): *Daucus aureus* Desf. (2*n* = 22; PI 319403), *D. conchitae* Greuter (2*n* = 22; Ames 25835), *D. carota* subsp. *capillifolius* (Gilli) C. Arbizu (2*n* = 18; PI 279764), *D. carota* subsp. *sativus* Hoffm. (2*n* = 18; DH1, a doubled haploid orange Nantes-type carrot), *D. carota* subsp. *sativus* (2*n* = 18; 'Dolanka'), *D. glochidiatus* (Labill.) Fisch & C.A. Mey (2*n* = 44; PI 285038), *D. guttatus* Sm. (2*n* = 20; PI 652233), *D. involucratus* Sm. (2*n* = 22; PI 652332), *D. littoralis* Sm. (2*n* = 20; PI 295857), *D. muricatus* (L.) L. (2*n* = 22; PI 295863), *D. pusillus* Michx. (2*n* = 22; PI 349267), *D. rouyi* Spalik & Reduron (2*n* = 20; PI 674284), *D. sahariensis* Murb. (2*n* = 18; Ames 29096), *D. syrticus* Murb. (2*n* = 18; Ames 29108), *Caucalis platycarpos* L. (2*n* = 20; PI 649446), *Orlaya daucoides* (L.) Greuter (2*n* = 16; PI 649477), *O. daucorlaya* Murb. (2*n* = 14; PI 649478), and *Torilis arvensis* (Huds.) Link (2*n* = 12; PI 649391).

### 2.2. Fruit Morphology

To characterize fruit morphology, 50 dry mericarps of each accession were placed on graph paper and photographed with a Flexacam C1 digital camera (Leica Microsystems, Heerbrugg, Switzerland) under a Leica S6D stereomicroscope (Leica Microsystems). The images were processed using Leica Application Suite X (Leica Microsystems) software, and the mericarp length (L) and width (W) were measured using AxioVision 4.8.2 software (Carl Zeiss MicroImaging, Jena, Germany). The fruit shape was described on the basis of the mean ratio of the mericarp length to width (L/W), and the following shape classes were used: ovoid (L/W ≤ 1.5), ellipsoid (L/W = 1.6–2.0), and narrowly ellipsoid or oblong (L/W ≥ 2.0), according to Lee et al. [40] and Mustafina et al. [41].

For scanning electron microscopy (SEM) analysis, dry fruit samples were mounted on stubs and sputter-coated with gold using a JFC-1100E ion sputter coater (JEOL, Tokyo, Japan); then, the dorsal side of the mericarps was examined under a JSM-5410 scanning electron microscope with a wolfram cathode (JEOL). The terminology used to describe the fruit surface sculpturing pattern was adopted from Stearn [42] and Ostroumova [43].

The fruit weight of each accession was expressed as grams per 100 mericarps and estimated by weighing four subsamples (each containing 50 randomly selected mericarps) using a WPS 510/C analytical balance (Radwag, Radom, Poland). The mean value was then calculated to obtain the weight of 100 mericarps.

### 2.3. Fruit Anatomy

For anatomical examination, 5–10 fruit samples (schizocarps or individual mericarps) of each accession were rehydrated in distilled water for 24–48 h, fixed in freshly prepared FAA (formalin, glacial acetic acid, and 70% ethanol, 6:4:90, $v/v/v$) for 48–72 h at room temperature, and stored in 70% ethanol at 4 °C until further use. The samples were then dehydrated in a graded ethanol series (80% and 90% for 2 h each) and left overnight in absolute ethanol. The dehydrated material was embedded in Technovit® 7100 resin (Kulzer, Hanau, Germany), following the manufacturer's protocol, with minor modifications involving prolonged infiltration with embedding solutions, i.e., the material was treated with increasing concentrations of Technovit relative to ethanol (1:3, 1:1, 3:1, $v/v$) for 24 h each and then left in pure Technovit for 5 days. The fixation, dehydration, and infiltration steps were performed on an orbital shaker (150 rpm) at room temperature, with 15 min vacuum pumping during each solution change. When polymerized, cross-sections of 4–8 μm thickness were made using a Leica RM2145 rotary microtome (Leica Microsystems, Wetzlar, Germany) with a Leica TC-65 carbide blade (Leica Microsystems). The sections were then stained with 0.2% ($w/v$) toluidine blue O (Sigma-Aldrich, Steinheim, Germany) for 30–60 s, mounted in Entellan® (Merck, Darmstadt, Germany), and analyzed under an Axio Imager.M2 microscope (Carl Zeiss, Göttingen, Germany). Three quantitative anatomical characteristics were measured (from five mericarps per accession): width of commissural vittae, width of vallecular vittae, and pericarp thickness. The terminology used to describe fruit anatomy follows that of Kljuykov et al. [22,23] and Wojewódzka et al. [35].

Another fruit sample was rehydrated in distilled water for 24 h and hand-sectioned using a disposable razor blade. The sections were then photographed under a stereomicroscope with the same camera as described in Section 2.2.

The transverse section of an exemplary mericarp showing the anatomical structures considered in this study, along with their terminology, is given in Figure 1.

### 2.4. Statistical Analysis

For each accession, the means and standard errors (SE) of the means were calculated for the measured quantitative parameters and then subjected to a one-way analysis of variance (ANOVA), followed by Tukey's honestly significant difference (HSD) test using Statistica 13.3 software (TIBCO Software Inc., Palo Alto, CA, USA). The differences were considered significant at $p \leq 0.05$.

## 3. Results

### 3.1. Fruit Morphology

The fruits of the studied taxa were schizocarps consisting of two homomorphic mericarps. The mericarps were pale yellow to brown in color and ovoid to oblong in shape in dorsal view (Figure 2 and Table 1). All taxa had spiny fruits, except for *D. rouyi* (Figure 2l), whose fruits were winged; however, since the material was mostly obtained from gene bank collections, the fruits often had broken spikes/wings or were devoid of these structures. In almost all taxa, the primary ribs were more or less inconspicuous or rarely prominent, covered with hairs or pointed thorns, whereas the secondary ribs (two dorsal and two

lateral) were remarkably prominent (Figure 2). *Torilis arvensis*, however, had numerous additional secondary ribs covering almost the entire surface of the fruit (Figure 2r).

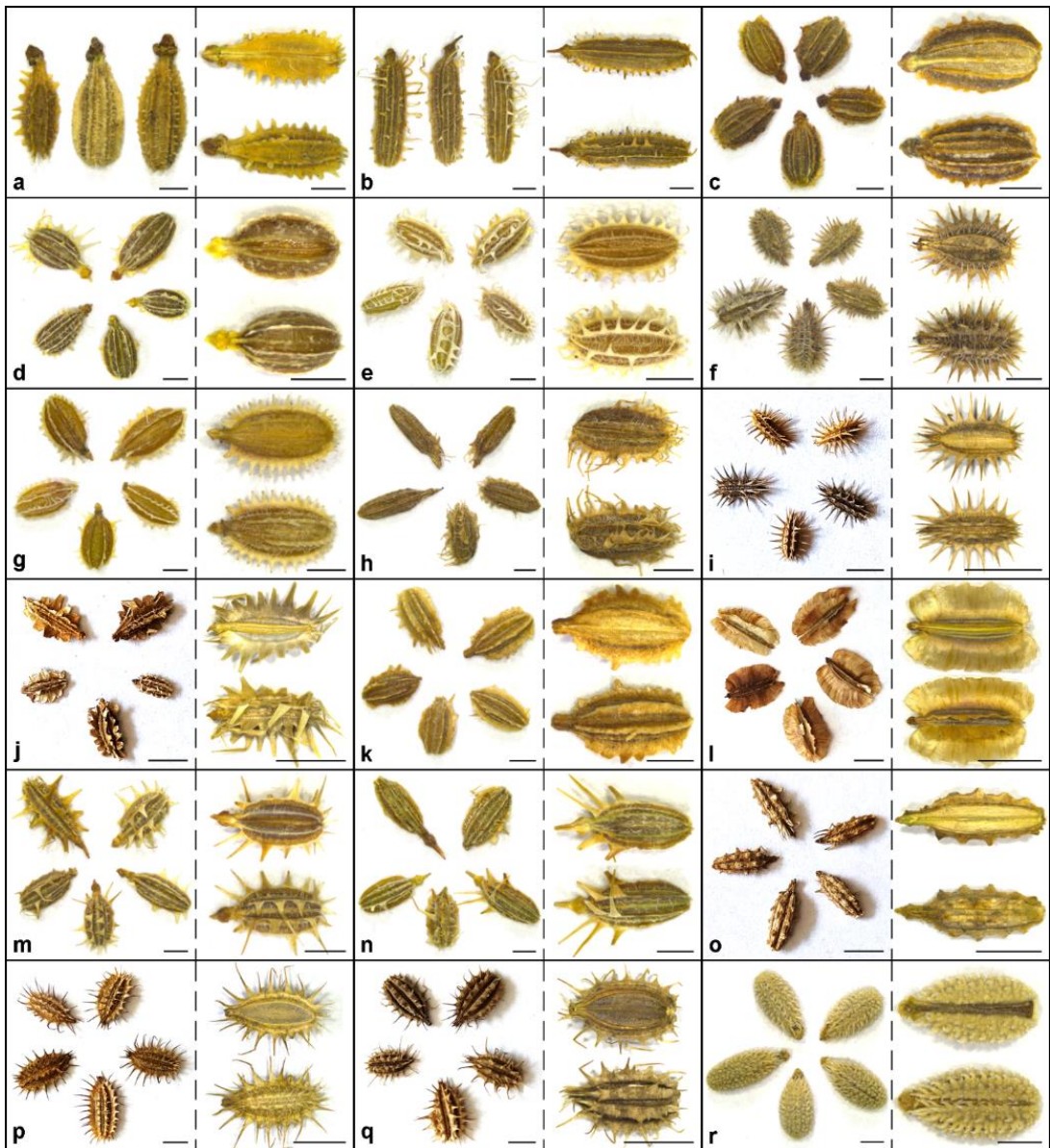

**Figure 2.** Variation in fruit morphology of the investigated *Daucus* and closely related non-*Daucus* taxa. The insets show magnified views of the dorsal (lower) and ventral (upper) sides of the mericarps. (**a**) *D. aureus*; (**b**) *D. carota* subsp. *capillifolius*; (**c**) subsp. *sativus* (DH); (**d**) subsp. *sativus* ('Dolanka'); (**e**) *D. conchitae*; (**f**) *D. glochidiatus*; (**g**) *D. guttatus*; (**h**) *D. involucratus*; (**i**) *D. littoralis*; (**j**) *D. muricatus*; (**k**) *D. pusillus*; (**l**) *D. rouyi*; (**m**) *D. sahariensis*; (**n**) *D. syrticus*; (**o**) *Caucalis platycarpos*; (**p**) *Orlaya daucoides*; (**q**) *O. daucorlaya*; (**r**) *Torilis arvensis*. Scale bars: 1 mm (**a–h,k,m,n,r**); 5 mm (**i,j,l,o–q**).

The mean mericarp length (L) varied from 2.1 (*D. carota* subsp. *sativus* 'Dolanka') to 11.4 mm (*O. daucoides*), whereas the average mericarp width (W) ranged from 1.1 (*D. conchitae* and *D. involucratus*) to 7.7 mm (*D. rouyi*) (Table 1). The ratio of these two parameters (L/W) was recorded in the range between 1.1 (*D. rouyi*) and 3.6 (*D. carota* subsp. *capillifolius*).

A closer look at the dorsal side of the mericarps, as examined under SEM (Figure 3), showed that the vast majority of *Daucus* taxa exhibited more or less rugose fruit surface sculpturing (Figure 3b–f,h,k,l). The most distinct pattern was found in *D. aureus* (Figure 3a), whose whole fruit surface was densely covered with tubercles (tuberculate type of sculpturing). Moreover, a few other or mixed types were detected. In *D. guttatus*, a rugose–tuberculate pattern was observed, i.e., rugose in the furrows between ribs, tuberculate on the surface of the secondary ribs (Figure 3g). *Daucus littoralis* showed a lineolate–tuberculate (lineolate furrows and tuberculate secondary ribs) surface (Figure 3i), whereas *D. rouyi* displayed ribbed–striate sculpturing (Figure 3j). In *D. muricatus*, the furrows were not clearly seen, but the surface of the secondary ribs was tuberculate (Figure 3m).

Among the outgroup species, variations in the types of sculpturing were also observed. In *T. arvensis*, the secondary ribs were densely covered with pointed tubercles (Figure 3n); *Caucalis platycarpos* exhibited a smooth surface (Figure 3o), while both *O. daucoides* and *O. daucorlaya* showed an undulate sculpturing pattern (Figure 3p,q, respectively).

In all cases, the outlines of the exocarp cells were not visible.

The lowest mean weight of 100 fruits (mericarps) was recorded for *D. glochidiatus* (0.079 g/100 fruits) and *D. syrticus* (0.080 g/100 fruits) (Table 1). *Orlaya daucorlaya* and *O. daucoides* had the heaviest fruits (3.451 and 3.407 g/100 fruits, respectively).

**Table 1.** Fruit (mericarp) morphological characteristics of the investigated *Daucus* and closely related non-*Daucus* (outgroup) taxa.

| Taxon | Length (L; mm) | | Width (W; mm) | | L/W | Shape | 100 Fruit Weight (g) |
|---|---|---|---|---|---|---|---|
| | Min–Max | Mean ± SE | Min–Max | Mean ± SE | | | Mean ± SE |
| ***Daucus*I subclade** | | | | | | | |
| *D. aureus* | 2.5–4.4 | 3.3 ± 0.06 h | 1.2–2.1 | 1.5 ± 0.03 de | 2.2 | NE | 0.136 ± 0.002 f–h |
| *D. carota* subsp. *capillifolius* | 4.0–6.5 | 4.9 ± 0.07 g | 1.1–1.7 | 1.4 ± 0.02 d–f | 3.6 | OB | 0.206 ± 0.002 f |
| *D. carota* subsp. *sativus* (DH) | 2.3–3.4 | 2.8 ± 0.04 i | 1.5–1.7 | 1.6 ± 0.01 d | 1.7 | E | 0.117 ± 0.003 f–h |
| *D. carota* subsp. *sativus* ('Dolanka') | 1.6–2.8 | 2.1 ± 0.04 k | 0.9–1.7 | 1.3 ± 0.02 fg | 1.7 | E | 0.139 ± 0.005 f–h |
| *D. muricatus* | 4.5–8.4 | 6.5 ± 0.12 e | 2.1–4.0 | 2.6 ± 0.05 c | 2.5 | NE | 1.076 ± 0.020 d |
| *D. rouyi* | 6.8–12.1 | 8.4 ± 0.13 c | 5.6–10.9 | 7.7 ± 0.15 a | 1.1 | OV | 1.349 ± 0.012 c |
| *D. sahariensis* | 2.0–3.8 | 2.7 ± 0.05 i | 0.9–1.6 | 1.3 ± 0.03 e–g | 2.1 | NE | 0.098 ± 0.002 gh |
| *D. syrticus* | 1.9–3.7 | 2.7 ± 0.06 i | 0.9–1.7 | 1.2 ± 0.03 fg | 2.2 | NE | 0.080 ± 0.002 h |
| ***Daucus*II subclade** | | | | | | | |
| *D. conchitae* | 1.9–3.7 | 2.5 ± 0.05 i–k | 0.8–1.5 | 1.1 ± 0.03 fg | 2.2 | NE | 0.106 ± 0.003 gh |
| *D. glochidiatus* | 1.8–3.0 | 2.2 ± 0.04 jk | 0.9–1.5 | 1.2 ± 0.02 fg | 1.9 | E | 0.079 ± 0.004 h |
| *D. guttatus* | 2.2–3.8 | 2.8 ± 0.05 i | 1.1–1.8 | 1.4 ± 0.03 d–f | 2.1 | NE | 0.109 ± 0.001 f–h |
| *D. involucratus* | 2.5–3.4 | 2.9 ± 0.03 i | 0.9–1.6 | 1.1 ± 0.02 g | 2.6 | OB | 0.098 ± 0.002 gh |
| *D. littoralis* | 4.8–6.7 | 5.7 ± 0.07 f | 2.1–3.3 | 2.6 ± 0.04 c | 2.3 | NE | 0.596 ± 0.011 e |
| *D. pusillus* | 2.0–2.9 | 2.5 ± 0.03 ij | 1.1–1.6 | 1.3 ± 0.02 e–g | 2.0 | NE | 0.091 ± 0.002 gh |
| **Outgroup** | | | | | | | |
| *Caucalis platycarpos* | 5.8–8.2 | 7.1 ± 0.07 d | 2.3–3.5 | 2.8 ± 0.03 c | 2.6 | OB | 1.664 ± 0.023 b |
| *Orlaya daucoides* | 8.7–14.0 | 11.4 ± 0.18 a | 4.2–8.1 | 5.8 ± 0.11 b | 2.0 | NE | 3.407 ± 0.061 a |
| *O. daucorlaya* | 6.8–12.5 | 10.1 ± 0.18 b | 3.8–7.3 | 5.8 ± 0.09 b | 1.8 | E | 3.451 ± 0.035 a |
| *Torilis arvensis* | 2.2–3.6 | 2.7 ± 0.03 i | 1.2–1.8 | 1.4 ± 0.02 d–g | 2.0 | NE | 0.178 ± 0.005 fg |

Means followed by the same letter in a column were not significantly different at $p \leq 0.05$. E, ellipsoid; NE, narrowly ellipsoid; OB, oblong; OV, ovoid; SE, standard error.

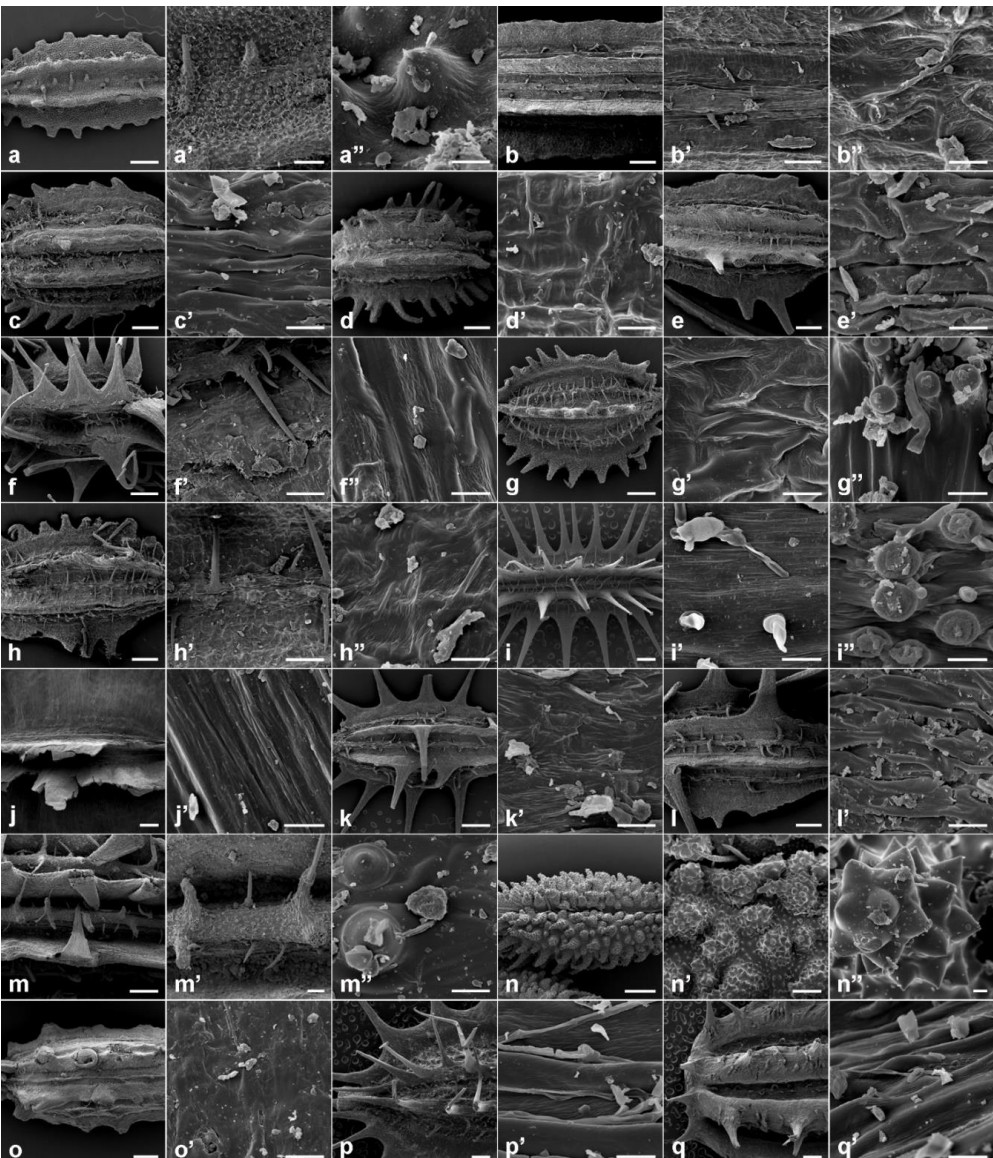

**Figure 3.** Fruit morphology and its surface micromorphology of the investigated *Daucus* and closely related non-*Daucus* taxa by scanning electron microscopy. (**a**–**a″**) *D. aureus*: (**a**) dorsal view and close-ups on (**a′**) the median primary rib and (**a″**) tubercle; (**b**–**b″**) *D. carota* subsp. *capillifolius*: (**b**) dorsal view and close-ups on (**b′**) the median primary rib and (**b″**) surface between ribs; (**c,c′**) subsp. *sativus* (DH): (**c**) dorsal view, (**c′**) close-up on the surface between ribs; (**d**–**d′**) subsp. *sativus* ('Dolanka'): (**d**) dorsal view, (**d′**) close-up on the surface between ribs; (**e,e′**) *D. pusillus*: (**e**) dorsal view, (**e′**) close-up on the surface between ribs; (**f**–**f″**) *D. conchitae*: (**f**) dorsal view and close-ups on (**f′**) the primary rib and (**f″**) surface between ribs; (**g**–**g″**) *D. guttatus*: (**g**) dorsal view and close-ups on (**g′**) the surface between ribs and (**g″**) tubercles; (**h**–**h″**) *D. involucratus*: (**h**) dorsal view and close-ups on (**h′**) the median primary rib and (**h″**) surface between ribs; (**i**–**i″**) *D. littoralis*: (**i**) dorsal view and close-ups on (**i′**) the surface between ribs and (**i″**) tubercles; (**j,j′**) *D. rouyi*: (**j**) dorsal view, (**j′**) close-up on the surface of the wing; (**k,k′**) *D. sahariensis*: (**k**) dorsal view, (**k′**) close-up on the surface between ribs; (**l,l′**) *D. syrticus*: (**l**) dorsal view, (**l′**) close-up on the surface between ribs; (**m**–**m″**) *D. muricatus*: (**m**) dorsal view and close-ups on (**m′**) the primary rib and (**m″**) tubercles; (**n**–**n″**) *Torilis arvensis*: (**n**) dorsal view and close-ups on (**n′**) the additional secondary ribs and (**n″**) tubercle; (**o**–**o′**) *Caucalis platycarpos*: (**o**) dorsal view, (**o′**) close-up on the surface between ribs; (**p,p′**) *Orlaya daucoides*: (**p**) dorsal view, (**p′**) close-up on the surface between ribs; (**q,q′**) *O. daucorlaya*: (**q**) dorsal view, (**q′**) close-up on the surface between ribs. Scale bars: 600 μm (**i,j,o**–**q**); 400 μm (**a,f,g,k,m,n**); 300 μm (**b**–**e,h,l**); 100 μm (**a′,b′,f′,h′,m′,n′**); 10 μm (**c′**–**e′,g′,i′**–**l′,o′**–**q′,a″,b″,f″**–**i″,m″,n″**).

### 3.2. Fruit Anatomy

The mericarp outline in the transverse section of almost all examined taxa was slightly compressed dorsally (Figures 4–6 and Table 2), except for *C. platycarpos*, which was slightly compressed laterally (Figure 6d).

Although the primary ribs of *Daucus* fruits were not prominent compared to the secondary ones (Figures 3–6), they were distinctly large in *D. muricatus* but still not larger than the secondary ribs (Figures 3m and 6c). The primary ribs were more or less similar in size and shape, whereas the secondary ribs often differed, with lateral secondary ribs usually longer than dorsal ones. The number of ribs in the mericarps was typically constant among taxa, i.e., five primary and four secondary, except for *D. littoralis*, in which mericarps with one additional primary and one additional secondary rib were rarely found (Figure 7a). Among the outgroup, the rib architectural pattern was similar to *Daucus*, i.e., more or less inconspicuous primary ribs and prominent secondary ribs; however, some distinct features of the latter were observed. The secondary ribs of *C. platycarpos* were wide and thick, often with a sunken apex (Figure 6d). *Orlaya daucoides* sometimes had bifurcated secondary ribs (Figure 6f), whereas those of *O. daucorlaya* were massive and thick, often clavate-shaped, with a thin base (Figure 6g). The fruits of *T. arvensis* were characterized by the presence of numerous additional secondary ribs (Figures 3n and 6e).

**Table 2.** Fruit (mericarp) anatomical characteristics of the investigated *Daucus* and closely related non-*Daucus* (outgroup) taxa.

| Taxon | Width (µm) | | Pericarp Thickness (µm) | Mericarp Outline [a] | Exocarp [b] | Hypendocarp | Endosperm [c] | Surface Micromorphology |
|---|---|---|---|---|---|---|---|---|
| | VV | CV | | | | | | |
| ***Daucus*I subclade** | | | | | | | | |
| *D. aureus* | absent | absent | 51 ± 5 c–e | SCD | T | – | F/C | Tuberculate |
| *D. carota* subsp. *capillifolius* | 139 ± 6 bc | 181 ± 16 cd | 28 ± 1 e | SCD | – | – | F/C | Rugose |
| *D. carota* subsp. *sativus* (DH) | 91 ± 4 ef | 115 ± 6 e–g | 38 ± 1 e | SCD | – | – | F/C | Rugose |
| *D. carota* subsp. *sativus* ('Dolanka') | 82 ± 8 fg | 78 ± 3 f–h | 32 ± 3 e | SCD | – | – | F/C | Rugose |
| *D. muricatus* | 33 ± 2 i | 83 ± 6 f–h | 108 ± 7 ab | SCD | – | – | F/C | Tuberculate |
| *D. rouyi* | 168 ± 7 a | 200 ± 7 c | 132 ± 9 a | SCD | – | – | F/C | Ribbed–striate |
| *D. sahariensis* | 75 ± 4 f–h | 122 ± 5 e–g | 42 ± 5 e | SCD | A | – | F/C | Rugose |
| *D. syrticus* | 70 ± 3 f–h | 144 ± 14 de | 48 ± 4 de | SCD | – | – | F/C | Rugose |
| ***Daucus*II subclade** | | | | | | | | |
| *D. conchitae* | 64 ± 3 gh | 71 ± 5 gh | 35 ± 2 e | SCD | – | – | F/C | Rugose |
| *D. glochidiatus* | 53 ± 3 hi | 53 ± 3 h | 34 ± 6 e | SCD | A | – | F/C | N/A |
| *D. guttatus* | 87 ± 4 e–g | 108 ± 8 e–g | 51 ± 4 c–e | SCD | – | – | F/C | Rugose–tuberculate |
| *D. involucratus* | 75 ± 2 f–h | 81 ± 2 f–h | 31 ± 2 e | SCD | – | – | F/C | Rugose |
| *D. littoralis* | 110 ± 5 de | 127 ± 8 ef | 80 ± 5 bc | SCD | – | – | F/C | Lineolate–tuberculate |
| *D. pusillus* | 85 ± 2 e–g | 125 ± 5 ef | 38 ± 5 e | SCD | – | – | F/C | Rugose |
| **Outgroup** | | | | | | | | |
| *Caucalis platycarpos* | 93 ± 2 ef | 87 ± 3 f–h | 117 ± 7 a | SCL | – | – | MG | Smooth |
| *Orlaya daucoides* | 125 ± 5 cd | 273 ± 17 b | 129 ± 6 a | SCD | – | + | F/C | Undulate |
| *O. daucorlaya* | 150 ± 12 ab | 329 ± 26 a | 118 ± 13 a | SCD | – | + | F/C | Undulate |
| *Torilis arvensis* | 81 ± 3 fg | 106 ± 7 e–g | 77 ± 6 cd | SCD | T | – | MG | Tuberculate |

[a] Mericarp outline in transverse section. [b] The presence or absence of exocarp protuberances or appendages. [c] Endosperm shape at commissure. Means followed by the same letter in a column were not significantly different at $p \leq 0.05$. A, cells with triangular appendages; CV, width of commissural vittae; F/C, flat or more or less concave; MG, mushroom-like grooved; N/A, not analyzed; SCD, slightly compressed dorsally; SCL, slightly compressed laterally; SE, standard error; T, covered with tubercles; VV, width of vallecular vittae.

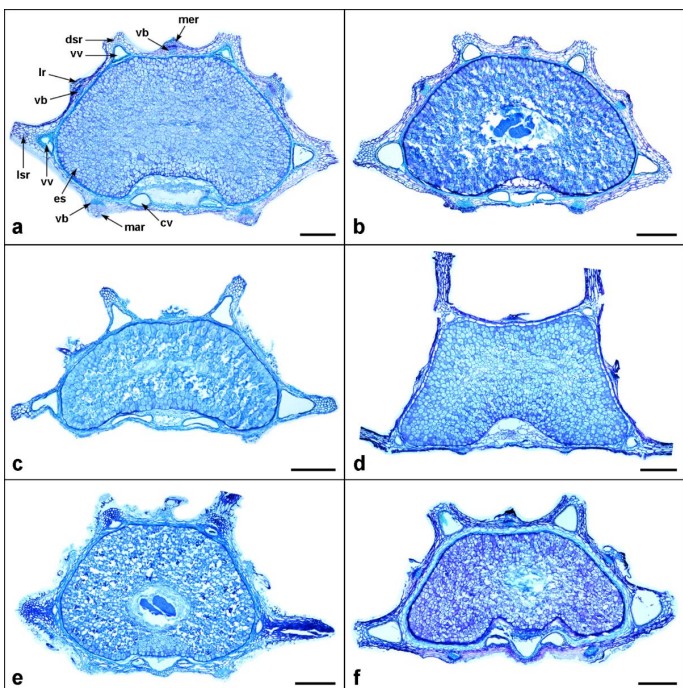

**Figure 4.** Mericarp structure of the investigated *Daucus* taxa, as seen in a transverse section. (**a**) *D. carota* subsp. *sativus* (DH); (**b**) subsp. *sativus* ('Dolanka'); (**c**) subsp. *capillifolius*; (**d**) *D. conchitae*; (**e**) *D. glochidiatus*; (**f**) *D. guttatus*. Abbreviations: cv, commissural vitta; dsr, dorsal secondary rib; es, endosperm; lr, lateral primary rib; lsr, lateral secondary rib; mar, marginal primary rib; mer, median primary rib; vb, vascular bundle; vv, vallecular vitta. Scale bar = 200 μm.

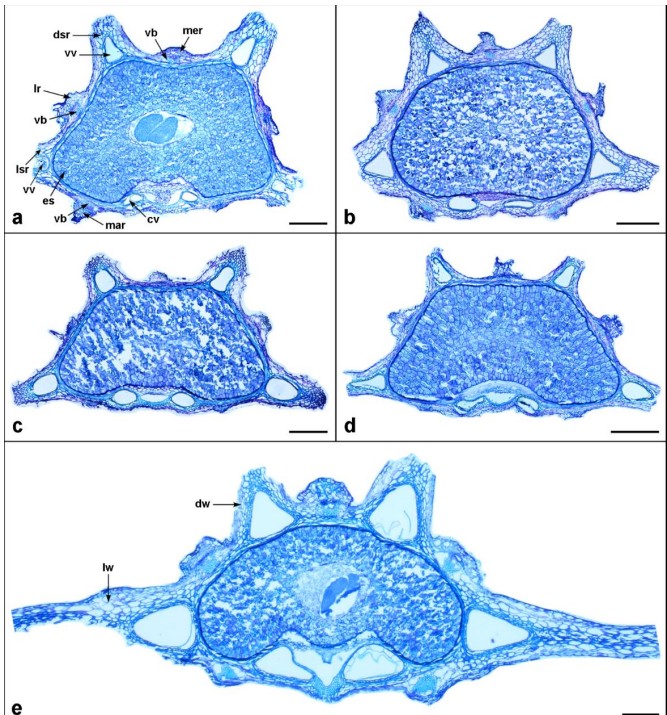

**Figure 5.** Mericarp structure of the investigated *Daucus* taxa, as seen in a transverse section. (**a**) *D. involucratus*; (**b**) *D. pusillus*; (**c**) *D. sahariensis*; (**d**) *D. syrticus*; (**e**) *D. rouyi*. Abbreviations: cv, commissural vitta; dsr, dorsal secondary rib; dw, dorsal wing; es, endosperm; lr, lateral primary rib; lsr, lateral secondary rib; lw, lateral wing; mar, marginal primary rib; mer, median primary rib; vb, vascular bundle; vv, vallecular vitta. Scale bar = 200 μm.

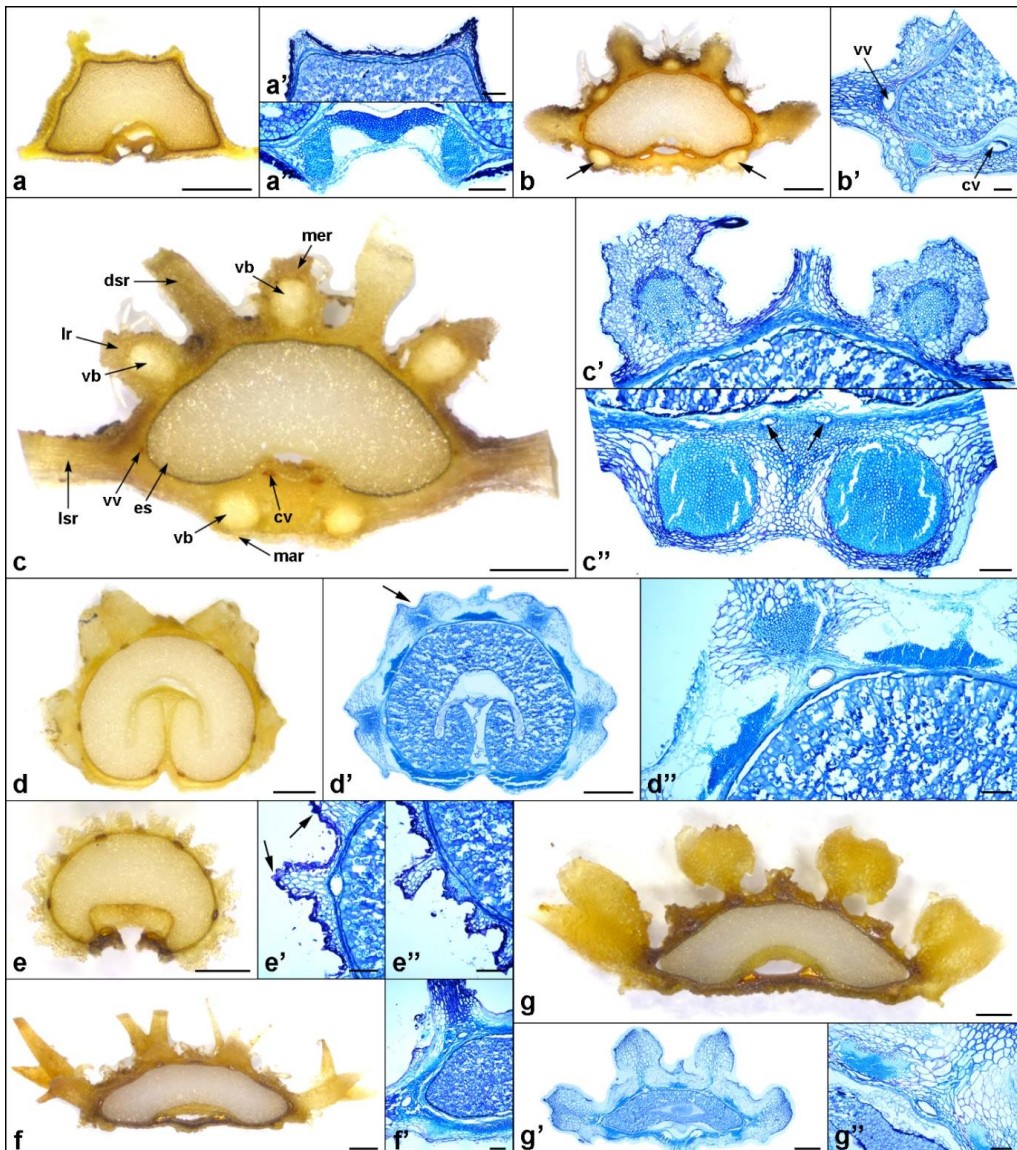

**Figure 6.** Mericarp structure of the investigated *Daucus* and closely related non-*Daucus* taxa, as seen in a transverse section. (**a–a″**) *D. aureus*; (**a′**) the upper part of the mericarp showing the absence of the vallecular vittae; (**a″**) M-shaped vascular bundle at the commissural side; (**b,b′**) *D. littoralis*, arrows indicate the larger vascular bundles in the marginal primary ribs; (**c–c″**) *D. muricatus*; (**c′**) the upper part of the mericarp showing two primary ribs with vascular bundles and the secondary rib in the middle enclosing the vallecular vitta; (**c″**) vascular bundles in the marginal primary ribs, arrows indicate commissural vittae; (**d–d″**) *Caucalis platycarpos*, arrow in (**d′**) indicates the sunken apex of the secondary rib; (**d″**) close-up on the upper part of the mericarp showing the flattened and elongated vascular bundles and a patch of collenchyma above the vallecular vitta; (**e–e″**) *Torilis arvensis*; (**e′**) close-up on the secondary rib enclosing the vallecular vitta, arrows indicate tubercles covering the exocarp; (**e″**) close-up on the part of the mericarp with secondary ribs and the primary rib in the middle; (**f,f′**) *Orlaya daucoides*; (**g–g″**) *O. daucorlaya*; (**g″**) close-up on the upper part of the mericarp showing the vasculature and the vallecular vitta. Abbreviations: cv, commissural vitta; dsr, dorsal secondary rib; es, endosperm; lr, lateral primary rib; lsr, lateral secondary rib; mar, marginal primary rib; mer, median primary rib; vb, vascular bundle; vv, vallecular vitta. Scale bars: 0.5 mm (**a–g,d′,g′**); 100 μm (**a′–c′,e′,f′,a″,c″–e″,g″**).

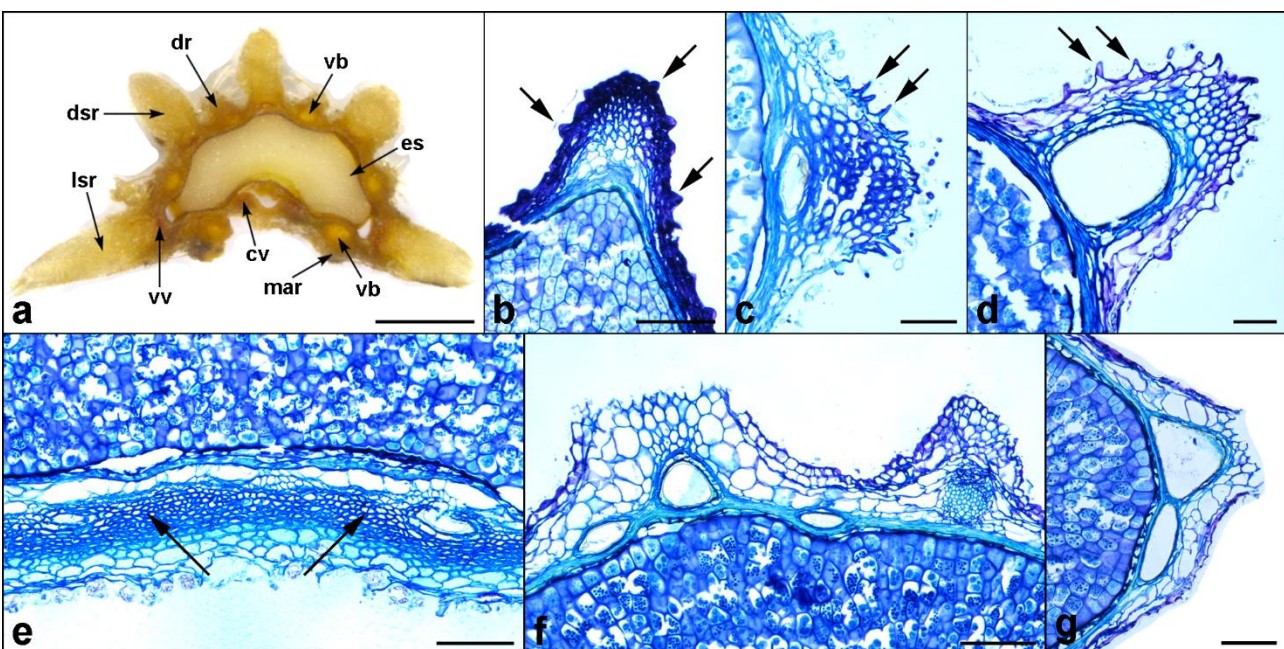

**Figure 7.** Selected distinct features or abnormalities found in the mericarps of *Daucus* and related taxa. (**a**) Abnormal mericarp of *D. littoralis* with additional dorsal primary and secondary ribs; (**b**) tubercles (arrows) on the exocarp of *D. aureus*; (**c**) characteristic exocarp cells with triangular appendages (arrows) covering the primary ribs of *D. glochidiatus* and (**d**) *D. sahariensis*; (**e**) close-up on the commissural side of *Orlaya daucoides* mericarps showing a hypendocarp (arrows); (**f**,**g**) additional smaller vallecular vittae in the cultivated carrot mericarps. Abbreviations: cv, commissural vitta; dr, dorsal primary rib; dsr, dorsal secondary rib; es, endosperm; lsr, lateral secondary rib; mar, marginal primary rib; vb, vascular bundle; vv, vallecular vitta. Scale bars: 1 mm (**a**); 50 μm (**c**,**d**); 100 μm (**b**,**e**–**g**).

The fruit wall (pericarp) of the investigated taxa had a typical structure of three layers: exocarp, mesocarp, and endocarp (see Figure 1), varying in thickness from 28 to 132 μm (Table 2). The single-layered exocarp consisted of small, thick-walled cells, usually flattened rectangular or more or less isodiametric in shape (Figure 1), but some exceptions were also found. In *D. aureus*, the exocarp was covered with numerous tubercles (Figures 3a and 7b), whereas, in *D. glochidiatus* and *D. sahariensis*, the part of the exocarp that covered the secondary ribs was composed of cells with triangular appendages (Figure 7c,d). The mesocarp consisted of a few to several layers of irregular thin-walled parenchymatic cells, typically larger than the exocarpic cells (Figure 1). The endocarp was a single compressed layer of somewhat lignified cells that usually adhered tightly to the seed coat (Figure 1). Regarding deviations in the pericarp structure among the outgroup taxa, we observed that the exocarp of *T. arvensis* was covered with numerous tubercles (Figures 3n and 6e), whereas the fruits of both *Orlaya* species were characterized by the presence of a hypendocarp, i.e., the inner fibrous mesocarp consisting of several layers of lignified fibers (Figure 7e).

Vallecular vittae were typically well developed in most members of *Daucus* and were triangular or ovate in shape (Figures 4–6); only *D. aureus* was devoid of these structures (Figure 6a). Among all taxa, *D. rouyi* exhibited the largest vallecular vittae (168 μm), whereas *D. muricatus* had the smallest (33 μm) (Table 2). The largest commissural vittae were found in *O. daucorlaya* (329 μm), and the smallest were found in *D. glochidiatus* (53 μm). Generally, each secondary rib enclosed one vitta; however, some variations were observed in carrots in which one or two additional smaller vittae—alongside the larger ones—were sometimes noticed (Figure 7f,g). All taxa, except for *D. aureus*, always had two commissural vittae that were ovate or compressed ovate in shape. Among the outgroups, the number and arrangement of both vallecular and commissural vittae were the same as in *Daucus*.

All *Daucus* taxa had a single compact vascular bundle embedded in the mesocarp below each primary rib. In *D. aureus*, however, the vasculature in the marginal primary ribs was connected in the commissure, forming a distinct M-shaped vascular bundle (Figure 6a). The size of the vascular bundles was more or less similar between the ribs of a given accession, except for *D. littoralis*, whose vascular bundles in the marginal primary ribs were distinctly larger than those in the dorsal primary ribs (Figure 6b). Among the outgroups, the most distinct differences in vasculature were flattened and elongated vascular bundles in the primary ribs of *C. platycarpos* (Figure 6d); the fruits of this taxon were also characterized by the presence of collenchyma in the secondary ribs.

In almost all taxa, the endosperm (commissural side) was flat or more or less concave, except for *C. platycarpos* (Figure 6d) and *T. arvensis* (Figure 6e), whose endosperm was mushroom-like grooved; *C. platycarpos* had strongly revolute margins.

## 4. Discussion

Traditionally, the taxonomic classification of the family Apiaceae has relied on the morpho-anatomical features of the fruits. However, many of the relationships inferred from this approach appear to be incongruent when confronted with molecular evidence. This is due to the high level of homoplasy among the fruit characteristics, which can be partially explained by selection [44]. Generally morphological characteristics are greatly affected by environmental factors [45–47]. Nevertheless, fruit characteristics can still provide useful information to support or supplement conclusions drawn from molecular data [27,32,44,48–50].

Here, we explored the morphology and anatomy of fruits in 13 *Daucus* and four closely related non-*Daucus* taxa. The results revealed a wide range of variation across the investigated taxa in terms of fruit size, shape, and weight, as well as fruit surface sculpturing and some anatomical characteristics. Thus, we pointed out several diagnostically valuable features of some of the *Daucus* taxa that we discuss below.

The morphometric characteristics and weights of the fruits differed significantly among the taxa (Table 1), which can be helpful—to some extent—in distinguishing between them. However, intra(sub)specific variations may occur in this regard, as observed here for cultivated carrot accessions. Moreover, in many cases, the quantitative values overlapped, which makes these data of limited taxonomic value. Therefore, the micromorphological features of the fruit surface, as well as fruit anatomy, appear to be more advantageous for distinguishing species.

Exocarp cell shape, exomesocarp protuberances, and cuticles are those components that contribute to fruit surface sculpturing, often providing taxonomically useful data [51]. In our study, as revealed by SEM, most of the investigated *Daucus* taxa had a rugose type of ornamentation, which can also be found, for instance, in *Ferula dshizakensis* [41] or some species of *Pimpinella* [52]. In *D. rouyi*, ribbed–striate fruit surface ornamentation was observed. This sculpturing pattern has also been reported, for example, in a few members of *Grammosciadium* [53] and *Pimpinella ibradiensis* [52]. Four *Daucus* taxa were characterized by the presence of tubercles, of which only *D. aureus* was covered on the entire surface of the mericarp, whereas *D. guttatus*, *D. littoralis*, and *D. muricatus* had only tuberculate secondary ribs. As for the exocarp cell shape (not visible by SEM), only *D. glochidiatus* and *D. sahariensis* were marked by the presence of distinct exocarp cells with triangular appendages that covered the surface of the secondary ribs; cells of this shape are characteristic of, for example, *Alepidea serrata* var. *serrata* [51]. Nonetheless, although the micromorphological characteristics of fruit surfaces have proven to be of taxonomic value, the application of these traits is difficult due to the lack of generally accepted terminology [54].

Species of *Daucus* and *Orlaya* (subtribe Daucinae), as well as *Caucalis* and *Torilis* (Torilidinae), are characterized by the presence of prominent secondary ribs, which is an almost unique trait among the members of these two subtribes and the genus *Artedia* [35,50]. In *Daucus*, the secondary ribs form spines or wings, the presence of which is a distinct adaptation to seed dispersal by epizoochory (animal-mediated dispersal) or anemochory

(wind-mediated dispersal). The genus *Daucus* has traditionally comprised only spiny-fruited species [5]; however, following a recent taxonomic revision by Banasiak et al. [11], numerous species with winged or obsolete fruits have been included in the genus. However, fruit appendages are characterized as highly homoplastic and are, thus, of limited utility in delimiting monophyletic groups [11,15].

The number and arrangement of both vallecular and commissural vittae within the pericarp are often of great taxonomic importance in Apiaceae. These secretory canals, located also in roots, stems, and leaves, are responsible for the specific odors of Apiaceae species as they contain essential oils, mucilage, gums, or resins [1], some of which are toxic to insects [55]. In our study, all taxa but one (*D. aureus*) had six vittae per mericarp: one below each secondary rib and two in the commissure, which is a common feature in most genera of Daucinae and Torilidinae [25]. Although we observed some variations in this regard in the cultivated carrot accessions that rarely had additional smaller vittae, these were presumably dwarf vittae, which could also be found, for instance, in *Apium graveolens* [56] or in many members of the Heteromorpheae tribe [57]. However, the size of the vittae seems to be more useful since this feature varied between many taxa.

In *Daucus* and related taxa, each mericarp had five vascular bundles—three in the dorsal primary ribs and two in the marginal primary ribs—as in almost all other members of Apiaceae. However, some exceptions to this pattern were found, for instance, in *Choritaenia capensis* [58] or *Cryptotaenia canadensis* [59], characterized by having seven vascular bundles, of which five were located on the dorsal side and two on the commissural side of the mericarp.

A lignified endocarp, composed of one layer of compressed and elongated cells, was present in all of the investigated *Daucus* taxa. De Miranda et al. [60] evidenced the process of lignin deposition in the endocarp cells of carrot fruit, along with their development, and reported that this process begins 21 days after anthesis.

Although the results showed considerable variations in the fruit morpho-anatomical characteristics, these variations were not sufficient enough to distinguish all of the investigated taxa. Exclusively on the basis of fruit characteristics, the most easily distinguishable taxon among *Daucus* was *D. aureus*, as it was characterized by several unique traits, i.e., entirely tuberculate fruit surface, lack of vittae, and distinct M-shaped vascular bundle on the commissural side. The partially tuberculate taxa (*D. guttatus*, *D. littoralis*, and *D. muricatus*) were distinguished by the length and weight of the mericarps, as well as by the features of their vascular bundles. The two taxa with characteristic exocarp cells with triangular appendages (*D. glochidiatus* and *D. sahariensis*) were differentiated according to the size of their vittae. In the case of *D. carota* subspecies, *D. carota* subsp. *capillifolius* differed from carrot accessions (subsp. *sativus*) by means of its mericarp length and oblong shape. *Daucus rouyi* was the only wing-fruited taxon in our sample. The remaining *Daucus* taxa (*D. syrticus*, *D. conchitae*, *D. involucratus*, *D. pusillus*, and the cultivated carrot) were morphologically and anatomically very similar to each other; thus, we were unable to unambiguously separate them.

## 5. Conclusions

This study provides detailed information on the morphology and anatomy of fruits from 13 *Daucus* and four closely related non-*Daucus* taxa. The results showed a wide range of variation in the fruit morpho-anatomical characteristics across the investigated taxa, as well as revealed several diagnostically valuable features of the fruits. For *Daucus*, the observed differences included the fruit size, shape (from ellipsoid to oblong), and weight, as well as the fruit surface sculpturing and some anatomical characteristics, i.e., the presence/absence and size of vittae, pericarp thickness, and the shape of exocarp cells. This study broadens the knowledge of the fruits of *Daucus* and may be useful for future taxonomical research on the genus and its close relatives.

However, to gain better insight into the relationships among the genus *Daucus*, further studies with a broader sample, including the remaining members of the genus, are needed.

**Author Contributions:** Conceptualization, D.K.; methodology, D.K.; formal analysis, D.K.; investigation, D.K.; resources, D.K. and E.G.; data curation, D.K.; writing—original draft preparation, D.K.; writing—review and editing, D.K. and E.G.; visualization, D.K.; supervision, E.G.; project administration, D.K. and E.G.; funding acquisition, D.K. All authors have read and agreed to the published version of the manuscript.

**Funding:** This research was funded by the National Science Center, Poland (grant number UMO-2019/35/N/NZ9/00959).

**Institutional Review Board Statement:** Not applicable.

**Data Availability Statement:** The data presented in this study are available in this article.

**Conflicts of Interest:** The authors declare no conflict of interest.

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
