# Peer review of "Comparative Fruit Morphology and Anatomy of Wild Relatives of Carrot (Daucus, Apiaceae)"

_agriculture, doi:10.3390/agriculture12122104_

Round 1
Reviewer 1 Report
1. What is the main question addressed by the research?
The authors want to answer;
What is differences between Daucus taxa and closely related non-Daucus species in terms of fruit morphology and anatomy?
They have answered this question, but there are some weaknesses, you can see my comments below.
2. Do you consider the topic original or relevant in the field?
Does it address a specific gap in the field?
Fruit morphological and anatomical characteristics are essential in the taxonomy of Apiaceae. The taxonomic and phylogenetic relationships among Daucus species have not yet been fully clarified. Therefore, morphological and anatomical investigations of fruits of Daucus taxa and closely related non-Daucus species are important to determine taxanomy of the genus. This study is addressing a specific gap in taxonomy of the Apiaceae family.
3. What does it add to the subject area compared with other published
material?
The subject provides an important resource for future studies on the Daucus genus and related genera. The photos of fruit morphology, detailed fruit anatomy photos and scanning electron microscopy (sem) photos reveal the fruit structural characteristics of Daucus taxa. It also reveals fruit differences between closely related non-Daucus species.
4. What specific improvements should the authors consider regarding the
methodology? What further controls should be considered?
- Authors should write the names of the authors where the plant names are written first in the text.
-Authors should write why they are examining 13 taxa of Daucus and four related non-Daucus species. Have they studied the taxa of a region?
-The authors can write detailed morphology of species, fruit surfaces and etc.
- The photos the authors provide for anatomy in the article are well, but when describing fruit anatomy they need to visually illustrate the differences in species, it could be a table or a cluster dendrogram.
5. Are the conclusions consistent with the evidence and arguments presented
and do they address the main question posed?
The author can briefly write the results of the study, and why these results are important.
6. Are the references appropriate?
References in the article are appropriate.
7. Please include any additional comments on the tables and figures.
Including figures showing general fruit characteristics in figure 1 provides convenience while reading the article. The photos of fruit morphology and fruit anatomy taken under a light microscope, and the sem photos show the characteristics of fruits.
The characteristics in Table 1 help us to understand the morphological differences between fruits. However, as I wrote above to understand the differences between species in terms of fruit anatomy, it is better to make a table or make a cluster dendrogram.
You can see some of the shortcomings I saw in the article in the comments in the attached pdf document.

Author Response
We thank the Reviewer for the evaluation of our manuscript and for all suggestions. We have revised the manuscript accordingly. Below, you can find detailed responses to the issues raised both in the comments and in the attached PDF document.
Authors should write the names of the authors where the plant names are written first in the text.
RESPONSE: As suggested by the Reviewer, the authors of the taxa have been added to the “Materials and Methods” section [lines 104–114].
Authors should write why they are examining 13 taxa of Daucus and four related non-Daucus species. Have they studied the taxa of a region?
RESPONSE: This study was a direct continuation of our previous research in which we used the same taxa. They were selected so that they cover the two main Daucus subclades (Daucus I and Daucus II). Moreover, these were the same accessions (the same accession numbers) that have commonly been used in previous phylogenetic and taxonomic studies on the genus Daucus. In the section “Introduction” [lines 87–92], to improve clarity, we have rephrased this paragraph.
It is better to state the full name of PI in the text.
RESPONSE: “PI” stands for Plant Introduction; these numbers are permanent numbers assigned to germplasm accessions in the National Plant Germplasm System. We have expanded the abbreviation PI, as suggested by the Reviewer, and its full name is now located in the text before the list of taxa [lines 103–104].
The characteristics in Table 1 help us to understand the morphological differences between fruits. However, as I wrote above to understand the differences between species in terms of fruit anatomy, it is better to make a table or make a cluster dendrogram.
RESPONSE: We thank the Reviewer for this suggestion and we have provided a new table (Table 2) that illustrates the fruit anatomical characteristics of the investigated taxa. Moreover, some additional morphometric data (width of vittae, pericarp thickness) have also been included in this new table.
Are the conclusions consistent with the evidence and arguments presented and do they address the main question posed?
The author can briefly write the results of the study, and why these results are important.
RESPONSE: As suggested by the Reviewer, we have supplemented the “Conclusions” section with the most important results obtained and their significance [lines 457–466].

Reviewer 2 Report
Minor suggestions need to include as highlighted in the attachment

Author Response
We thank the Reviewer for the evaluation of our manuscript and for all suggestions. We have revised the manuscript accordingly. Below, you can find detailed responses to the issues raised both in the comments and in the attached PDF document.
Minor suggestions need to include as highlighted in the attachment.
RESPONSE: We greatly appreciate the feedback from the Reviewer.
In the “Abstract” section, we have added the word “Apiaceae” in the second sentence [line 9] and rephrased the forth sentence [line 13], as pointed out by the Reviewer.
Kindly enumerate which species has 16, 18… chromosomes.
RESPONSE: In our opinion, enumerating all the Daucus species along with their chromosome numbers does not seem to fit the “Introduction” section of the manuscript (there are about 40 species). Therefore, we have supplemented the “Materials and Methods” section and provided the information on chromosome numbers of the taxa analyzed in this study [lines 103–114].
Pictorial representation of these would be more significance to support this text, with relevance to the botany of seed.
RESPONSE: We strongly agree that the description of fruit characteristics of Apiaceae would be clearer for readers when graphically presented. That is why we have prepared Figure 1 (“Materials and Methods” section), which depicts the most important features of the fruit. We have referenced this figure in the fragment underlined by the Reviewer in the “Introduction” section [line 58].

Reviewer 3 Report
Fruit morphology has been used significantly in the literature to help solve the complexity of Daucus taxonomy. The article brings valuable additional information and observations on Daucus seeds, especially through image analysis on whole fruits and also transversal sections of mericarps. The authors propose some features that can be useful to distinguish between Daucus species or taxa. However, they conclude that not all accessions can be identified based on these features, even if some differences are observed for some of them.
With these limitations and the lack of data on image analysis, the article appears quite speculative. The title should be revised as there are no “taxonomic implications” presented or discussed. Similarly, the conclusion should be revised as there is no better understanding provided or discussed about the “relationships among the genus Daucus”. In relation to that, the sentence in line 389-390 is not convincing and undermine the value of the work (recognize some of them).
Actually many observations of quality are provided with various features on a set of diverse and complementary accessions. However, it is quite difficult to synthetize the information to help distinguish among accessions. A synthetic table with the main features or a kind of determination key would be useful to valorize the results.
The literature is quite extensive and well presented. However, it should also include Mezghani et al (2014) on Daucus fruit descriptors.
The original aspect of the work is the image description and analysis of mericarps morphology and anatomy. Data are provided on width and length of fruits, but no data are provided from microscopy and transverse sections, only qualitative observations. Measurements are possible and may help to support some observations, for example related to mesocarps or vittae.
Some specific questions
- What is the origin of fruits : from wild or cultivated conditions in Amesof , what are the production conditions (same year, agronomic conditions?). This is important as the conditions in which the mother plants have been grown have big consequences on the features of the fruits, and may explain some variation if not grown in the same conditions
- Line 127: the 5 to 10 samples used were for technical reasons of also to observe variations?
- Line 163-164: why daucus and non daucus were computed separately, as the objective was to distinguish all accessions from the others and between clades.
- Figure 2: the distinction between dorsal and ventral is not always obvious. Is there a mistake of bar size in ventral pictures, as it is sometimes bigger or smaller compared to the dorsal pictures?
- Line 185: please rewrite “differed significantly” or be more specific, as there are no significant differences between many accessions and between subclades I and II. Even some Outgroup accessions are not significantly different from the Daucus accessions (please include them in the whole anova analysis).
- Figure 3: could you please indicate more information and legend below the title. The pictures of a, b, c (etc) do not seem to be the same aspect of the mericarp. Also, we don’ see what need to be observed/understood from pictures ‘ and ‘’, and actually it differs from one picture to another (sometimes spines, others not clear…). Please make sure that this figure can be understood independently from the text.
- Line 223: same comments as in line 185
- Figures 5, 6 and 7: please add the legend as in fig 4, to be better understandable (figures need to be independently understood)
- Lines 280-284: could measurements be made (and statitics) to support the conclusions
- Lines 323-328 : the incongruence with molecular data is common, as the morphologic data are influenced by the environment and correspond to a very limited portions of the genome.
- Line 368 etc: vittae deserve more discussion as they may be important, and there are some contradictions between line 370 (almost constant) and line 392 (absence, distinct shape). Some image analysis with measurements on vittae are feasible and could help with clear conclusions. The interest of vittae to understand Daucus evolution could be more discussed.
Author Response
We thank the Reviewer for the evaluation of our manuscript and for all suggestions. We have revised the manuscript accordingly. Below, you can find detailed responses to the issues raised in the comments.
Fruit morphology has been used significantly in the literature to help solve the complexity of Daucus taxonomy. The article brings valuable additional information and observations on Daucus seeds, especially through image analysis on whole fruits and also transversal sections of mericarps. The authors propose some features that can be useful to distinguish between Daucus species or taxa. However, they conclude that not all accessions can be identified based on these features, even if some differences are observed for some of them.
With these limitations and the lack of data on image analysis, the article appears quite speculative. The title should be revised as there are no “taxonomic implications” presented or discussed. Similarly, the conclusion should be revised as there is no better understanding provided or discussed about the “relationships among the genus Daucus”. In relation to that, the sentence in line 389–390 is not convincing and undermine the value of the work (recognize some of them).
RESPONSE: We appreciate the Reviewer's insightful comments on the taxonomical aspects as they pointed out that we had overinterpreted the results. As suggested, we have revised the title by removing the phrase “(…) and their taxonomic implications” and rephrased the “Discussion” [lines 442–443] and “Conclusions” section [lines 658–460, 467–469].
Actually many observations of quality are provided with various features on a set of diverse and complementary accessions. However, it is quite difficult to synthetize the information to help distinguish among accessions. A synthetic table with the main features or a kind of determination key would be useful to valorize the results.
RESPONSE: As this was also a suggestion of the Reviewer 1, we have provided another table (Table 2) that summarizes the fruit anatomical characteristics of the investigated taxa.
The literature is quite extensive and well presented. However, it should also include Mezghani et al. (2014) on Daucus fruit descriptors.
RESPONSE: We thank for this suggestion. We have included the following reference in the “Introduction” section [lines 65–67]. The reference numbering and reference list have been updated accordingly.
Mezghani, N.; Zaouali, I.; Bel Amri, W.; Rouz, S.; Simon, P.W.; Hannachi, C.; Ghrabi, Z.; Neffati, M.; Bouzbida, B.; Spooner, D.M. Fruit morphological descriptors as a tool for discrimination of Daucus L. germplasm. Genet. Resour. Crop Evol. 2014, 61, 499–510. https://doi.org/10.1007/s10722-013-0053-6
The original aspect of the work is the image description and analysis of mericarps morphology and anatomy. Data are provided on width and length of fruits, but no data are provided from microscopy and transverse sections, only qualitative observations. Measurements are possible and may help to support some observations, for example related to mesocarps or vittae.
RESPONSE: We thank the Reviewer for pointing it out. As suggested, we have provided the morphometric data on the width of commissural vittae, width of vallecular vittae, as well as the pericarp thickness. These data have also been subjected to the statistical analysis and included in the new Table 2. The sections “Materials and Methods” [lines 152–154] and “Results” [lines 326–327, 340–355] have been updated accordingly.
Some specific questions:
– What is the origin of fruits: from wild or cultivated conditions in Ames of, what are the production conditions (same year, agronomic conditions?). This is important as the conditions in which the mother plants have been grown have big consequences on the features of the fruits, and may explain some variation if not grown in the same conditions.
RESPONSE: Although the fruit samples were ordered from the USDA-ARS North Central Regional Plant Introduction Station (Ames, Iowa, USA) in the same year, we were not provided with details on the maintenance of these accessions in the gene bank. However, each accession is assigned with a specific method of growing; thus, the growing conditions during routine regeneration and maintenance procedures should remain the same.
– Line 127: the 5 to 10 samples used were for technical reasons of also to observe variations?
RESPONSE: Since the plant material was provided from the gene bank, the number of fruits were limited. Thus, we used 5 to 10 fruits as biological replications to evaluate variation among analysed taxa as the members of the genus are outcrossing.
– Line 163–164: why Daucus and non Daucus were computed separately, as the objective was to distinguish all accessions from the others and between clades.
– Line 185: please rewrite “differed significantly” or be more specific, as there are no significant differences between many accessions and between subclades I and II. Even some Outgroup accessions are not significantly different from the Daucus accessions (please include them in the whole anova analysis).
– Line 223: same comments as in line 185
RESPONSE (to the three comments above): As reasonably suggested by the Reviewer, we have computed the statistical analyses including all the studied taxa (Daucus and non-Daucus taxa) and modified the Table 1. The “Materials and Methods” section [line 174–175] and information below Table 1 [line 207–209] have been updated accordingly. As also suggested, we have rephrased the above-mentioned paragraphs [lines 197–204 and 248–253].
– Figure 2: the distinction between dorsal and ventral is not always obvious. Is there a mistake of bar size in ventral pictures, as it is sometimes bigger or smaller compared to the dorsal pictures?
RESPONSE: Thank you very much for catching it! Indeed, there is a mistake here – the scale bars in the insets of Figure 2 (i, j, l, o, p, and q) should show 5 mm instead of 1 mm. We have changed these scale bars and provided the modified Figure 2.
– Figure 3: could you please indicate more information and legend below the title. The pictures of a, b, c (etc) do not seem to be the same aspect of the mericarp. Also, we don’ see what need to be observed/understood from pictures ‘ and ‘’, and actually it differs from one picture to another (sometimes spines, others not clear…). Please make sure that this figure can be understood independently from the text.
RESPONSE: We have provided details to the legend of Figure 3, as requested by the Reviewer [lines 228–245].
– Figures 5, 6 and 7: please add the legend as in fig 4, to be better understandable (figures need to be independently understood).
RESPONSE: We thank for this suggestion. We have provided modified Figures 5, 6, and 7 and adjusted the legends accordingly [lines 267–270, 285–288, and 321–323].
– Lines 280–284: could measurements be made (and statistics) to support the conclusions
RESPONSE: It was difficult to precisely measure the single-layered exocarp, especially in the case of D. aureus whose external cell layers were too intensely stained, which made it difficult to unambiguously distinguish the cell layer boundaries. Thus, to avoid inaccuracies, we have removed this part of the sentence [line 329].
– Lines 323–328: the incongruence with molecular data is common, as the morphologic data are influenced by the environment and correspond to a very limited portions of the genome.
RESPONSE: Thank you, we have added this information to this paragraph [line 375–376].
– Line 368 etc: vittae deserve more discussion as they may be important, and there are some contradictions between line 370 (almost constant) and line 392 (absence, distinct shape). Some image analysis with measurements on vittae are feasible and could help with clear conclusions. The interest of vittae to understand Daucus evolution could be more discussed.
RESPONSE: As suggested by the Reviewer, we have rephrase this paragraph and provided some additional information to the “Results” [lines 340–355] and “Discussion” sections [lines 418–429].

Round 2
Reviewer 3 Report
Valuable improvement of the paper